# MAIN: A Real-world Multi-agent Indoor Navigation Benchmark for Cooperative Learning

**Fengda Zhu**[1]    **Siyi Hu**[1]    **Yi Zhang**[1]    **Haodong Hong**[1]
**Yi Zhu**[3]    **Xiaojun Chang**[1]    **Xiaodan Liang**[2,4]
[1]Monash University    [2] Sun Yat-sen University
[3] Huawei Noah's Ark Lab    [4] Dark Matter AI Inc.

## Abstract

The ability to cooperate and work as a team is one of the 'holy grail' goals of intelligent robots. Previous works have proposed many multi-agent reinforcement learning methods to study this problem in diverse multi-agent environments. However, these environments have two limitations, which make them unsuitable for real-world applications: 1) the agent observes clean and formatted data from the environment instead of perceiving the noisy observation by themselves from the first-person perspective; 2) large domain gap between the environment and the real world scenarios. In this paper, we propose a Multi-Agent Indoor Navigation (MAIN) benchmark[1], where agents navigate to reach goals in a 3D indoor room with realistic observation inputs. In the MAIN environment, each agent observes only a small part of a room via an embodied view. Less information is shared between their observations and the observations have large variance. Therefore, the agents must learn to cooperate with each other in exploration and communication to achieve accurate and efficient navigation. We collect a large-scale and challenging dataset to research on the MAIN benchmark. We examine various multi-agent methods based on current research works on our dataset. However, we find that the performances of current MARL methods does not improve by the increase of the agent amount. We find that communication is the key to addressing this complex real-world cooperative task. By Experimenting on four variants of communication models, we show that the model with recurrent communication mechanism achieves the best performance in solving MAIN.

## 1 Introduction

Cooperative multi-agent problems are ubiquitous in real-world applications, for example, multiplayer games [40, 38, 18], multi-robot control [29], language communication [48, 15, 33], and social dilemmas [23]. These applications focus on solving the sequential decision-making problem of multiple autonomous agents within a common environment, which could be systematically modeled as the multi-agent reinforcement learning (MARL) paradigm [34, 52, 65]. Compared to traditional reinforcement learning, MARL has two major challenges. The first is the partial observability. Each agent observes only part of the global state. The second is the instability of learning decentralised policies. Recent works have proposed diverse environments to validate the effectiveness of the MARL algorithms, such as grounded communication environment [33], StarCraft II [42], DOTA2 [7], multi-agent emergence environments [4], soccer shooting [26], etc.

Most of these game-based MARL environments are quite different from the real-world situation such as robotics and auto-driving. For example, the agent observes clean and formatted data from the

---

[1]http://main-dataset.github.io/

Submitted to the 35th Conference on Neural Information Processing Systems (NeurIPS 2021) Track on Datasets and Benchmarks. Do not distribute.

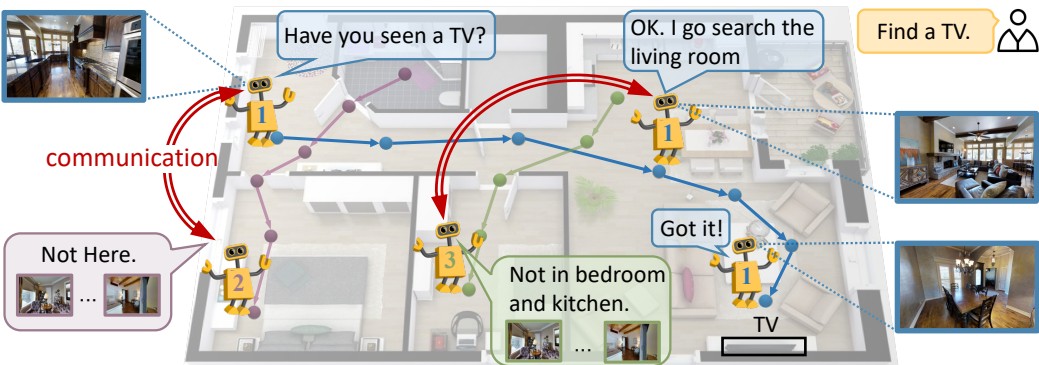

Figure 1: A demonstration of our Multi-agent Indoor Navigation (MAIN) benchmark. The agent 1 moves in the room to find the target by cooperating and communicating with agent 2 and agent 3.

game environment instead of perceiving realistic observations by themselves from the first-person perspective. The transition function in the game environments is based on simple rules without simulating the physical rules and considering the interference in the real world. Therefore, there is large domain gap between current MARL environments and real world, which limits current MARL models [27, 40, 62] to be applied to real-world scenarios.

*There are no good models without good data* [41]. To overcome these limitations, we propose a novel benchmark, Multi-Agent Indoor Navigation (MAIN), where multiple agents are required to navigate to reach goals in a 3D indoor room. To obtain realistic observations from the first-person view of agents, we adopt Habitat [44] simulator to render realistic egocentric RGB-D image observations for agents. At each navigation step, each agent observes an RGB-D image from its own first-person perspective and makes action decision including, 'turn left', 'turn right' and 'step forward'. The setting of realistic egocentric observation is closer to real-world situations, which makes the learned agents easier to be transferred to real-world applications like robotics.

In other MARL environments such as StarCraft II [42] and multi-agent emergence environments [4], the observations of different agents have a large proportion of overlapping. For example, the status (position or health) of an agent or an object is fully observed if it is located within the vision range of another agent. It is unrealistic compared to the real-world where the status of agents and objects is not fully observable. In our proposed MAIN environment, the appearance, shape, and size of an object will be very different, when observed by the agents from different angles, especially in the first-person views where the angles are dynamically changing all the time. In addition to the realistic observation, MAIN adopts the Bullet physics engine [12] to provide a more realistic transition function. Unlike other environments [42, 4, 26], where the agent receives the high-precision localization information from the environment, MAIN does not provide a compass sensor and requires the agent to navigate solely using an egocentric RGB-D camera. Compared with previous single-agent navigation environments such as MINOS [43] and Habitat [44], we implement a asynchronous-synchronous pipeline for efficient multi-agent data sampling.

To the best of our knowledge, our MAIN is the first multi-agent real-world navigation environment. The environments of MAIN benchmark bring new challenges, such as learning from realistic observations and less observation overlapping between agents. These new challenges raise additional requirement for better utilizing the information by sharing the individual observation with other collaborators and making decisions based on both self-observation and collaborators' observation. However, many MARL methods [40, 1, 18, 55] adopt the Centralized Training Decentralized Execution (CTDE) [36] framework, which forbids the real-time information sharing among agents. Therefore, these methods are not suitable for a real-world simulated environment like MAIN. To address this, we propose a new cooperative multi-agent communication mechanism to enable the agents to exchange the information in a real-time manner. This communication mechanism may not be essential for game-based or highly-simplified tasks like Hanabi [5], SMAC [53] and Hide-and-seek [4] but is crucial in real-world tasks like MAIN due to the highly unlinear and little overlapped observations. In short, being aware of the status of the collaborators is beneficial to making a good action decision, and is critical for accomplishing a real-world cooperative task. An overview of MAIN task with the communication mechanism is shown in Fig. 1. Three agents start from different

| Input | Environment | Multi-agent | Embodied | Observation Overlap | Physics Engine |
|---|---|---|---|---|---|
| Array | Hanabi [5] | ✓ | - | $\frac{N-1}{N}$ | - |
| | Diplomacy [37] | ✓ | - | $\frac{N-1}{N}$ | - |
| | EGCL [33] | ✓ | ✗ | 100% | - |
| | DOTA2 [7] | ✓ | ✗ | 100% | Rubikon |
| | SMAC [42] | ✓ | ✗ | 81.0%-89.8% | Havok |
| | Hide-and-seek [4] | ✓ | ✗ | $\frac{1}{N}$-100% | MuJoCo |
| | Soccer [26] | ✓ | ✗ | 100% | MuJoCo |
| Syn image | MARLÖ [39] | ✓ | ✓ | $\frac{1}{N}$ | hybrid voxel |
| | AI2-THOR [21] | ✗ | ✓ | $\frac{1}{N}$ | Unity |
| Real image | Habitat [44] | ✗ | ✓ | $\frac{1}{N}$ | Bullet |
| | MAIN (Ours) | ✓ | ✓ | $\frac{1}{N}$ | Bullet |

Table 1: Compared with existing MARL environments (N is the number of agents). The egocentric view means if the agent has an local observation of environment from its perspective rather than receiving global information. The embodied view represents whether an agent observe the 3D environment from the first-person perspective (we only compare the navigable environments).

positions and are asked to find a TV in this room. Each agent receives a first-person photo-realistic observation from their perspective respectively. They explore the room and communicate with each other to exchange their discoveries. By active exploration and cooperative communication, the No. 1 agent finally finds the TV.

Considering that agents can have different targets at the same time, we propose two sub-tasks for our MAIN task: 1) **Shared-target navigation** where all agents are asked to find a shared target $g$; 2) **Individual-target navigation** where each agent $i$ has its own target $g_i$. In the shared-target navigation sub-task, we mainly evaluate the agents cooperation ability of searching separately for a target. In the individual-target navigation, we focus on evaluating the ability of cooperative information exchanging. To fully investigate the MAIN benchmark, we construct a large-scale dataset consists of 24M episodes within 90 houses for training, validation, and testing splits, which is 10 times larger than the dataset in [44]. The data is automatically labeled within the environment. Compared with other datasets [57, 44], our dataset is challenging since it provides more long-term hard samples.

Along side with the environment, we provide multiple baselines for MARL research community for fast evaluating their effectiveness for real-world deployment. We build our baseline models based on previous MARL works [27, 1, 62] to validate the effectiveness of our benchmark and dataset. We find that the number of agents improves the navigation performance in simple baselines. However, without communication, the navigation performance will not increase by increasing the number of agents. And we find it is essential for agents to communicate with each other in addressing complex real-world cooperation tasks. We experiment on four kinds of communication variants and find that the model with a recurrent actor-critic mechanism to encode historical communication messages significantly outperforms other models. In summary, we make the following contributions: 1) we propose the MAIN benchmark to research on multi-agent problem in a realistic environment; 2) we collect a large-scale and challenging dataset and benchmark several MARL baseline models; 3) we propose a communication module that benefits for the real-world multi-agent system.

## 2  Related Work

**Multi-agent Environments** have been proposed to research multi-agent problems. However, previous works ignore the importance of implementing a realistic environment, which limits the learned model to be applied on real-world applications such as robotics. In Tab. 1, we compare the differences between our MAIN environment with previous multi-agent environments. To the best of our knowledge, we claim that our MAIN environment is the first multi-agent environment that offers realistic image input. Previous works [7, 42, 4] get clean and formatted array data via programming interfaces. Therefore, it would be hard for the learned model to overcome the challenge of the interference from noisy data. MARLÖ [39] provide synthetic image whose domain is largely deviated from the scenarios of the real-world.

Our model provides an egocentric and embodied view, which is quite applicable for robots in the real world. A particular challenge in the MARL problem is partial observability. Each agent could only observe a part of the global state and solve the problem by communication. However, our investigation reveals that some of the previous MARL environments [42, 4], even though claimed to be partially observable, have large observation overlap. Little observations overlap makes our benchmark more challenging than the previous environments, because little information sharing makes cooperation difficult. A realistic physical engine helps simulate a real-world transition function. Our environment adopt Bullet engine to simulate physics activities such as acceleration and collision.

**Multi-agent Reinforcement Learning** extends the problem of reinforcement learning [32, 20, 50] into multi-agent scenario and brings new challenges. Most MARL problems [28, 47, 65] falls into the centralized training with decentralized execution (CTDE) architecture [36, 22, 61]. Some works are dedicated to improving the mixing network which mixes the agent network outputs to learn a joint action-value function [40, 49, 55]. Other works are aiming at improving the network structure and developing a individual function with better representation and transfer capability [16, 18]. Besides, the utilization of state information varies. IPPO [13] incorporates the global state information barely by sharing network parameters among critics of individual agents. While MAPPO [63] constructs a centralised value function upon agents which takes the aggregated global state information as inputs. Researchers find that communication is critical cooperative multi-agent problems. Lowe *et al.* [27] propose MADDPG, a framework based on DDPG [24] with cooperative value function. Later, R-MADDPG [54] equips with recurrent actor crtic models, simultaneously learning policies for navigation and communication towards better information utilization and resource distribution. We implement diverse methods to support extensive research and illustrate the novelties and challenges of MAIN.

**Embodied Navigation Environments.** Simulations such as Matterport3D simulator [3], Gibson simulator [60] and Habitat [44] propose high-resolution photo-realistic panoramic view to simulate more realistic environment. Rendering frame rate is also important to embodied simulators since it is critical to training efficiency. MINOS [43] runs more than 100 frame per second (FPS), which is 10 times faster than its previous works. Habitat [44] runs more than 1000 FPS on $512 \times 512$ RGB+depth image, making it become the fastest simulator among existing simulators. Some complex tasks may require a robot to interact with objects, such as picking up a cup, moving a chair or opening a door. AI2-THOR [21], iGibson [59] and RoboTHOR [14] provide interactive environments to train such a skill. Multi-agent reinforcement learning [25, 51] is a rising problem of cooperation and competition among agents. Based on the Habitat simulator, we construct a multi-agent environment to research on realistic MARL problem.

**Embodied Navigation Learning** is attracting rising attention in the community and lots of methods have been proposed to address this problem. Based on conventional reinforcement learning methods [31], Wu *et al.* [58] introduce an LSTM layer to encode the historical information. Wang *et al.* [56] propose to jointly learn a navigation model with imitation learning and supervised learning. Some works [19, 30, 66] propose auxiliary tasks to exploit extra training signals for learning navigation. SLAM-based methods [64, 9, 10] are widely adopted in navigation due to its capability of modeling the room structure. Nonetheless, those tasks do not conducted in multi-agent setting which requires cooperation and communication, and consequently, being more flexible and practical.

## 3 Multi-agent Indoor Navigation Benchmark

### 3.1 Task Definition

Here we define our proposed Multi-agent Indoor Navigation (MAIN) Benchmark in detail. MAIN requires multiple agents $E = \{e_1, ..., e_n\}$ to navigate to reach a set of targets accurately and efficiently in an indoor environment. At the beginning of an episode, each agent is told to reach a target $g_i$. For each step, the agent observes an observation and make an action decision. The observation contains an RGB-D image, localization information from a GPS compass and contact information from a physics sensor. An action could be 'turn left', 'turn right', 'step forward' and 'found'. The agent uses the first three actions to navigate in the environment and uses the last action to declare it has found the target object. The episode of this agent is considered succeed if the 'found' action is selected while the agent is located with the threshold toward the target object. Otherwise, the episode is consider a

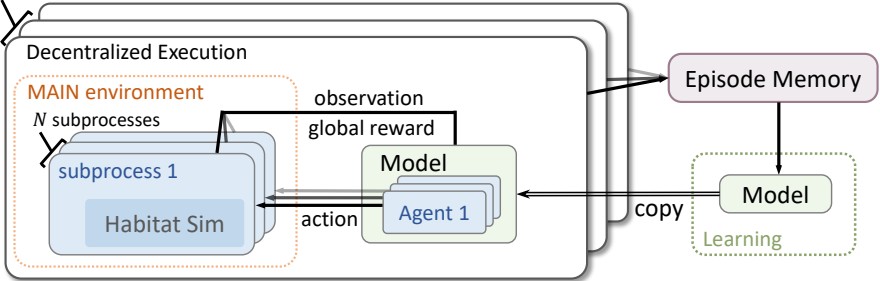

Figure 2: Training overview of our MAIN environment.

failure if the 'found' action is sleeted while the agent is out of range or it has navigated for maximum steps without finding the target.

Based on the above rules, we propose two sub-tasks, shared-target navigation and individual-target navigation. **Shared-target navigation** is a task where all agents are asked to find a shared target $g$. The task is considered succeed if any agent reaches $g$ and considered failure if any agent fails within its episode. **Individual-target navigation** is a task where each agent $i$ has its own target $g_i$, and the task is considered succeed only when all agents successfully find its own target. We do experiments under the setting of different agents and different tasks to demonstrate the challenge and novelty of this benchmark. The reward function is defined by shortened distance similar to [57].

## 3.2   Multi-agent Indoor Navigation Environment

This environment is built based on Habitat [44] simulator. Habitat simulator renders the 3D assets of an house and provide a photo-realistic embodied environment for agents. The Habitat simulator provide multiple sensors including RGB-D image, GPS compass and contact. The Habitat is built upon the Bullet physics engine that enables realistic graphics rendering, velocity and acceleration simulation, and contact simulation. However, the rendering process is computation consuming and time costly. Therefore, we design a asynchronous-synchronous pipeline for data efficiency.

Our pipeline is shown in Fig. 2. The MAIN environment creates $B$ sub-environments for decentralized execution to sample data for training, where $B$ is the size of the minibatch. Each sub-environment creates $N$ processes, where the $N$ is the number of agents. Each process has a copy of a Habitat simulator, and each simulator individually simulates the state of an agent and renders the RGB-D image observation for an agent. The MAIN sub-environment synchronizes the processes and interacts with a copy of a multi-agent navigation model. In the decentralized execution, the parameters of the the multi-agent navigation model are shared across all sub-environments. The multi-agent navigation model predicts actions for each agent for each step. The predicted actions are sent to the MAIN environment and then distributed to each process to execute. The Habitat simulator execute the action and return the updated state and the current partial observation to the MAIN sub-environment. The MAIN sub-environment calculate the global reward based on the global state and send the global reward and observations to the model. For each step, the global reward, observations for all agents and actions that agents predict are stored in the episode memory. We sample the episodes from the episode memory to optimize a centralized model by stochastic gradient descent (SGD). The model after a step of SGD optimization is copied to each MAIN sub-environment to update the execution model.

## 3.3   Data Collection

We use the room textures and other 3D assets provided by Matterport3D [8] to build the MAIN environment. Matterport3D consists of 10,800 panoramic views constructed from 194,400 RGB-D images of 90 building-scale scenes, where 61 scenes for training, 11 for validation, and 18 for testing following the standard split [8]. We provide episode data for learning and testing. An episode is defined by a starting position where the agent starts and the target position where the agent is required to reach. Both the starting position and the target positions are randomly sampled from navigable points within an environment. We ensures there is at least one navigable path from the starting

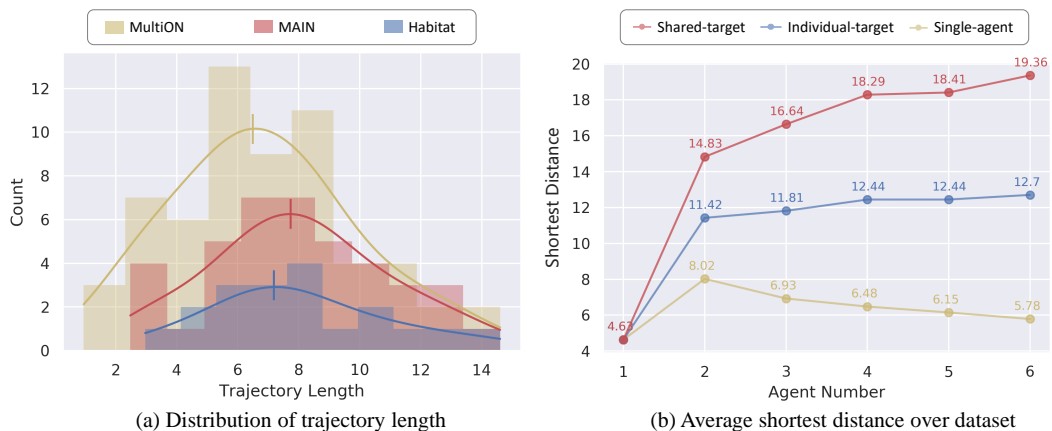

Figure 3: An analysis of our MAIN dataset. The curve in the left figure stands for the Gaussian smoothing. The lines are the mean values of the smoothed Gaussian distributions.

position to the target. And we constrain the length of an episode to be between 2m and 20m, which ensures that each episode is neither too trivial nor too hard.

We compare the distribution of the average trajectory length within a room with MultiON [57] and the object navigation data in the Habitat Challenge [6] in Fig. 3(a). Due to the different structures of the house, the episode data from each house have different average length. We find that with the same room setting, our average trajectory length is longer than both MultiON and Habitat, proving that our data is more challenging. Our dataset provide 24M episodes, 10 times more than the data scale of the Habitat dataset.

The Fig. 3(b) shows the average distance that the agents need to navigate to successfully accomplish task. It reveals the gap of the difficulty among the two sub-tasks and the single-agent navigation task accompany with the agent amount using our dataset. With the increase of the agent amounts, the difficulty of individual-target task is significantly increasing while the shared-target task is reducing.

### 3.4 Metrics

The MAIN task is evaluated from two aspects: navigation accuracy and efficiency. We use the following metrics to quantitatively measure the effectiveness of models:

**Success Rate** is used to measure if the agent successfully finds the target when it yields 'found'. The agent is regarded 'success' only if it is located within a threshold distance towards the target.

**Distance** indicates the average distance forward the target when the agent stops. This metric is useful when the success rate is low.

**SPL**, short for Success weighted by Path Length [2], evaluates the accuracy and efficiency simultaneously. The SPL is calculated by $\frac{1}{N} \sum_{i=1}^{N} S_i \frac{p_i}{l_i}$, where the $N$ is all testing samples, $S_i$ is the success indicator, $p_i$ is the shortest path length, and the $l_i$ is the actual path length in testing. We adopt the SPL as our main metric.

## 4 Multi-agent Models

### 4.1 Preliminaries

We systematically model our MAIN problem as a multi-agent reinforcement learning paradigm which is described as a partially-observed Markov decision process (POMDP) [35]. $P(s'|s, a)$ is the transition probability that transforms the current state space $S$ to the next state space $S'$ conditions on the a global action $a \in A$. We follow the centralized-training decentralized-execution framework that parameterize the shared policy of each agent as $\pi_\theta$. For each step $t$, the agent $i$ receive its partial observation $o_{t,i}$ and choose its action by $a_{t,i} = \pi_{\theta_i}(o_{t,i})$. The global action $a_t = a_{1,t}, ..., a_{n,t}$ All agents share the same global reward function $r(s, a) : S \times A \to \mathbb{R}$. And the $\gamma \in [0, 1)$ is a discount

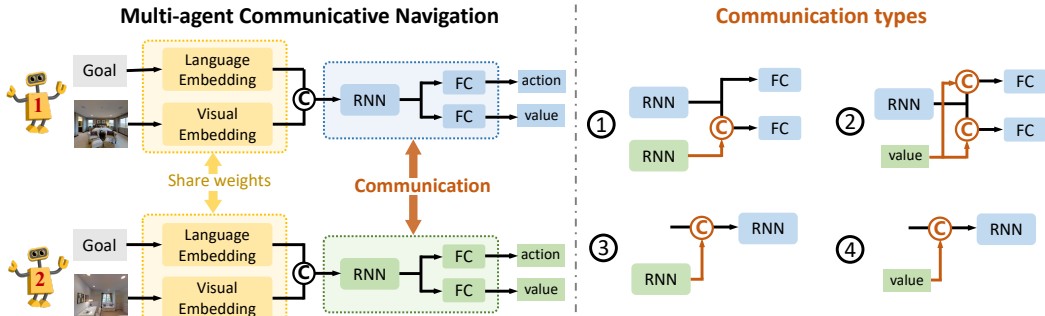

Figure 4: A demonstration of our multi-agent communicative navigation framework. We take a two-agent model for instance. A dot bounding box denote an agent. The dual-directed orange arrow stands for weight sharing between the orange boxes. The orange arrow stands for communication. On the right side, we show four kinds of communication variants.

factor that defines the length of the horizon. We optimization the parameter $\theta$ by minimizing the optimization objective $J(\theta) = \mathbb{E}\left[\sum_t \gamma^t r(s_t, a_t)\right]$ by PPO algorithm [46].

## 4.2  Baseline Multi-agent Models

We implement several multi-agent models to investigate the performance of multi-agent models on the MAIN task.

**Random navigator with oracle founder.** We implement a random baseline which randomly sample the action of 'turn left', 'turn right' and 'go forward'. And the baseline model has the oracle 'found' module that yields 'found' as long as the agent reaches within the success range of navigation. This baseline model is used to validate if our dataset is too easy or have severe bias.

**Multi-single agent.** This model is implement in the PPO [46] that is trained in a single-agent paradigm but tested in a multi-agent paradigm. We research on this model to see if the number of agents help the performance of navigaion in multi-agent paradigm.

**IPPO.** The IPPO model learns the global reward and share network parameters each agent. The difference between PPO and IPPO [1] is that the PPO model receives a single-agent reward while the IPPO model receives a global reward that influenced by other agents. The actions of other agents cause the instability of the global reward, which increases the difficulty of training.

**MAPPO.** Based on IPPO, MAPPO [63] introduces a centralised value function upon agents with global state inputs. However, the original MAPPO does not consider the importance of encoding the historical communicative information, which limits its application in complex environments where the observations of the agents have little in common and the historical information is important in action decision. In our implementation, the CNN and RNN are shared among agents while the each agent has its own actor and critic functions.

## 4.3  Multi-agent Cooperative Communication Navigation

In this section, we are going to introduce our cooperative communicative navigation model, as shown in Fig. 4. We take a two-agent situation for demonstration. The framework firstly embeds the target as an embedding feature, and extracts visual feature using an Convolutional Neural Network (CNN) [17] module. The parameters of the embedding layer and the CNN layer are shared between agents to ensure the generazability. Then the target feature and visual feature are concatenated to feed the Recurrent Neural Network (RNN) [11] module. The RNN module is adopted to encode historical information. Since the agents receive partial observation, it is important to memorize the previous observation to help the agent build a more comprehensive understanding of the environment. The historical feature from the RNN is send to two fully connected layers. One outputs an probability that represents the preference of making action decision and the other predicts a value to estimate the effective of the current situation. The model is optimized by PPO algorithm. To be specific, the action prediction is supervised by policy gradient loss and the value prediction is supervised by the bellman equation.

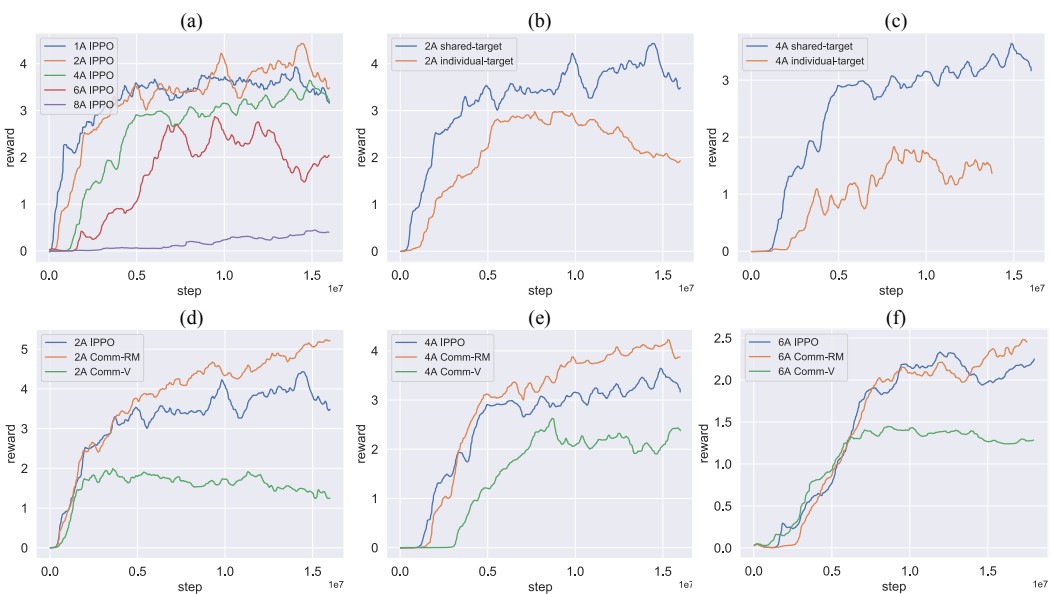

Figure 5: The result curves of our experiments.

| Models | 2 Agents | | | | 3 Agents | | | |
|--------|----------|----------|--------------|------|----------|----------|--------------|------|
| | Length | Distance | Success rate | SPL | Length | Distance | Success rate | SPL |
| Random | 3.39 | 12.75 | 0.00 | 0.00 | 3.30 | 16.67 | 0.00 | 0.00 |
| Multi-PPO [46] | 232.89 | 10.66 | 0.21 | 0.17 | 47.11 | 16.10 | 0.01 | 0.00 |
| IPPO [1] | 256.67 | 15.55 | 0.08 | 0.06 | 137.24 | 15.94 | 0.02 | 0.01 |
| Comm-S | 351.03 | **10.02** | 0.12 | 0.06 | 75.92 | 16.16 | 0.05 | 0.05 |
| Comm-V | 68.14 | 12.02 | 0.03 | 0.03 | 80.23 | **10.36** | 0.05 | 0.04 |
| Comm-RM | 309.7 | 12.78 | **0.3** | **0.23** | 312.86 | 14.59 | **0.13** | **0.06** |
| Comm-RV | 298.1 | 11.56 | 0.24 | 0.17 | 301.2 | 12.32 | 0.08 | 0.05 |

Table 2: The testing results of different models. Multi-PPO: single-agent PPO model tested in multi-agent environment. The four variants of our communicative models is denoted as Sequential Communication model (Comm-S), Value Communication model (Comm-V), recurrent message communication model (Comm-RM), and recurrent value communication model (Comm-RV).

The agents exchange information between the the blue block and the green block to obtain more knowledge and build a more comprehensive understanding of the environment. The feature vector that an agent send is named as 'message'. The agent that receives the message is named the 'receiver' and the agent that sends the message is named the 'sender'. The gradient is not back-propagated from the 'receiver' to the 'sender' since it causes severe instability in training, which makes the performance of the learned navigation model to be almost zero. On the left we show four communicative variants. We name the them as sequential communication model (Comm-S) , value communication model (Comm-V), recurrent message communication model (Comm-RM), and recurrent value communication model (Comm-RV).

## 5 Experiment

**Implementation Details** Our communicative model is built based on our implementation of [1]. We train all of our models for 15M iterations. We adopt Adam optimizer whose learning rate is $2.5 \times 10^{-4}$. The discount factor $\gamma = 0.99$ and the TD($\lambda$) factor in GAE [45] is 0.95. Our model is trained on by 8 GPUs (7 GPUs for rendering image inputs and 1 GPU for optimization) for 36 hours.

**Ablation for Agent Amount** The Fig. 5(a) ablates the amount of agent in MARL learning. We find that with the amount increasing, the navigation performance is declining. More agent narrows the searching area for find a target. However, the global reward is easily effected by the actions of other agents, and therefore, hard to give an agent a clear guidance.

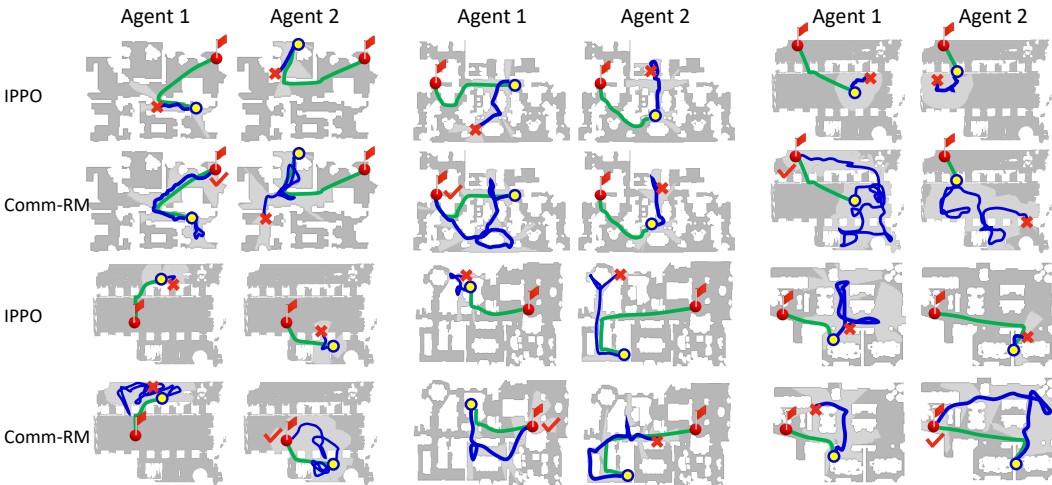

Figure 6: The trajectory visualization results of the IPPO agent and the communicative agent in the testing environment. The red circle with a red flag is the position where the target located. The yellow circle is the starting position of an agent. The green line indicates the shortest path, and the blue line is the actual navigation path. The red cross indicate the location where an agent fails.

**The Difficulty of Two Sub-tasks** The Fig. 5(b), (c) ablate the difficulty of two sub-tasks. We train the IPPO baselines on individual-target task and shared-target task respectively. We find that the individual-target task is significantly harder than the shared-target task, and the gap of difficulty is increasing with more agent amounts. This experimentation result also proves the dataset analysis result in Sec 3.3.

**Ablation for Communication** We train the model with historical communication mechanism, the model with value communication mechanism and the IPPO baseline on 2 agents, 4 agents and 6 agents scenario. The result if shown in the Fig. 5(d), (e), (f), where the model with historical communication mechanism significantly outperform other two models. In addition, we find that the value communication mechanism cause overfitting in the MAIN task.

A more detailed comparison is shown in Tab. 2. We test our baseline models and the Comm-RM model in both 2-agents and 3-agent scenarios. We find that the third variant, whose structure is shown in Fig. 4, performs the best and largely outperforms other methods. We conclude from this figure that communication mechanism is quite important for the MAIN task. A proper communication mechanism largely improves the performance while a bad design of cooperative mechanism may introduce noise or cause overfitting. Moreover, we find that the results of the IPPO model and the single-agent PPO model tested in the multi-agent environment still competitive.

**Visualization for Navigation Process** In Fig. 6, we visualize the navigation process of two models: the IPPO baseline model and the Comm-RM model. In this figure, at lease one agent from the communicative model successfully reaches the target. We find that the agents with cooperative communication is able to explore larger area and navigation for a longer trajectory. Similar result is also observed in Tab. 2. We find that the agents with communication tend to explore different areas in a room. It indicates that the agents is able to learn to navigate seperately and communicate the exploration result, which largely improve the navigation efficiency.

## 6 Conclusion

In this paper, we propose a novel Multi-Agent Indoor Navigation (MAIN) benchmark to research on multi-agent problem in a realistic environment. We collect a large-scale dataset for researching on MAIN and analysis the advantage of our dataset. We benchmark multiple baseline models in MAIN and find that traditional MARL methods cannot solve MAIN due to the unique challenges in MAIN such as little observation overlap and high variance of the embodied image view. By doing experimentation, We discover that the model with historical communication message significantly helps multi-agent navigation in MAIN. In the future, we are going to research on MARL problems based on MAIN and keep updating the dataset and the codebase of MAIN.

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
