# OpenReview forum: "MAIN: A Multi-agent Indoor Navigation Benchmark for Cooperative Learning"
_NeurIPS.cc/2021/Track/Datasets_and_Benchmarks/Round1 — Submitted to NeurIPS 2021 Datasets and Benchmarks Track (Round 1)_

### Official Review · Reviewer_YYCW · 2021-06-23
**A relevant problem, but benchmark, dataset and method lack significant details**

**Rating:** 3
**Confidence:** 4

**Strengths:**

This work refers to observation overlap as a significant challenge for MARL training (overlap as a degree of similarity between observations of multiple concurrently acting agents). This appears to be a relevant problem which to the best of my knowledge is not covered in prior work. Unfortunately, the problem is not further discussed beyond the introduction.

The challenge of egocentric visual input problems for robotics and similar domains is valid and appears convincing. I suggest to focus on these two motivational properties of MAIN to highlight differences to prior MARL environments (in contrast claims about partial observability appeared less convincing).

The difficulty of MAIN tasks and their correlation to the number of agents are nicely visualised and illustrated in Section 3.3 (but some analysis does not seem to hold up / lacks detail --> see below).

**Weaknesses:**

**Overall:**

The paper tries to provide a new environment, dataset, a benchmark (within the new environment) as well as a new communication method to address challenges within the environment. In doing so, significant details are missing for each part. I suggest a more focused approach restricted to the environment and dataset for instance which might help overall clarity. Following, I will provide more detailed feedback regarding the motivation as well as each of the four contributions.



**Motivation:**

The work largely motivates the need for multi-agent environments with egocentric image observations based on "easier (transfer) to real-world applications like robotics" (l. 47). While I agree with the statement in general, it remains unclear whether agents trained in MAIN are able to be transferred to real-world robots. Even sophisticated simulations are still simplifications of the real-world and MAIN incorporates high-level actions which do not match real-world actuation.

This potentially remaining challenge of sim-to-real gap for MAIN might diminish the motivated benefits over environments such as MALMO/MARLÖ [39] or also DeepMind Lab (1, see below) which use egocentric image observations in multi-agent environments based on video-games.



**MAIN environment:**

No significant details are provided regarding the environment interface. It would be helpful to precisely state observations (e.g. for RGB-D image which precise shape/ resolution), reward function for stated global reward and transition function. E.g. how far does the agent move and turn with a single such action?



Secondly, the "asynchronous-synchronous pipeline" appears very computationally expensive with N * B instances of the Habitat simulator running concurrently (1 simulator required for each agent in B parallel environments). In particular given the required rendering, this appears extremely costly. Comparison to computation cost of other MARL simulations might help to quantify this cost.





**Large-scale dataset:**

The paper provides very few information about the released dataset and even the supplementary material and webpage do not fill that gap. The dataset contains "episode data for learning and testing" (l. 201) but besides the mentioned starting and target positions it is unclear what information the dataset contains. Documentation about the content and format of the dataset is entirely missing in paper, supplementary material, webpage and even the downloaded dataset itself (which only includes compressed json files in a simple folder structure). It would be very helpful if you could state

- what the dataset contains (information for each timestep? Actions selected? Rewards/ observations? ...)
- in which format the dataset contains such information
- and ideally provide examples of how information can be extracted from the dataset.

I downloaded the dataset myself but given the large file size of these json-formatted files, it is very hard to identify the format myself. Good documentation is essential to make the dataset easily usable and adopted by the research community.

Lastly, it is not mentioned how the data has been collected. Did you collect this data during training of some algorithms? If so which were used? ...



**MAIN benchmark:**

A benchmark alongside a newly proposed environment is typically meant to to identify how current methods perform in the new set of tasks. However, the conducted benchmark suffers from missing details and coverage:

- All baselines and communication models are based on PPO. It would significantly strengthen any arguments if other MARL algorithms (e.g. for CTDE - QMIX (2), MADDPG (3), COMA (4) or Independent Learning - DQN (5), A2/3C (6)) were considered. Similarly, if a novel communication method is presented and communication is stated to address challenges of the MAIN environment, I would expect a comparison to related work in communication for MARL (e.g. 7 - 11).
- Results are reported without standard error/ std, confidence intervals or similar metrics which would allow reasoning over the significance of differences and evaluation over multiple random seeds is not mentioned. (MA)RL is known to be sensitive to random seeds, hyperparameters and implementation details (12), so evaluation over multiple seeds is essential for meaningful evaluation.
- For Figure 5 and Table 2, it is not clear in which MAIN tasks these results were generated.
- From Table 2, it appears Multi-PPO (Is this MAPPO of 4.2?) outperforms IPPO in 2 agents and 3 agents scenarios. I'd then expect comparisons to this stronger baseline in Figure 5 (d), (e) and (f).
- Contrary to the author's claim in Checklist 3. (d), the type of resources are not fully specified (I could not find such information in the main paper, supplementary material nor the provided webpage). Merely the number of GPUs is stated, but it would help to categorise computational cost with specific GPU models (and potentially further hardware specification) being stated.
- In l. 294f, it is claimed that experimentation results given by the performance in the two sub-tasks with 2 and 4 agents "proves the dataset analysis results in Section 3.3". However, in Section 3.3 and in particular Figure 3b it is shown that the shortest path is larger for the shared-target task and grows faster with an increase in the number of agents. This seems to suggest a larger increase in difficulty of the shared-target task with the number of agents compared to the individual-target task and appears contrary to the claim made here.





**Communication module:**

The communication module proposed in Section 4.3 is put in no context with related work on communication in MARL (e.g. 7 - 11). Also, it is not clearly motivated how you identified in particular communication as a promising approach in MAIN. Significant details are missing and only high-level descriptions of the conducted communication are provided. This section would significantly benefit from added formalism to clearly define the proposed method. Furthermore:

- Agents are stated to be categorised as "receiver" or "sender" agents for the proposed communication (l. 275f). Are these roles pre-defined or somehow learned? How are agents assigned to such roles?

- Four variants of the proposed communication model are mentioned but never explained (only visualised in Figure 4 right side) - l. 279ff.





**References:**

(1) Beattie, Charles, Joel Z. Leibo, Denis Teplyashin, Tom Ward, Marcus Wainwright, Heinrich Küttler, Andrew Lefrancq et al. "Deepmind lab." *arXiv preprint arXiv:1612.03801* (2016).

(2) Rashid, Tabish, Mikayel Samvelyan, Christian Schroeder, Gregory Farquhar, Jakob Foerster, and Shimon Whiteson. "Qmix: Monotonic value function factorisation for deep multi-agent reinforcement learning." In *International Conference on Machine Learning*, pp. 4295-4304. PMLR, 2018.

(3) Lowe, Ryan, Yi Wu, Aviv Tamar, Jean Harb, Pieter Abbeel, and Igor Mordatch. "Multi-agent actor-critic for mixed cooperative-competitive environments." *arXiv preprint arXiv:1706.02275* (2017).

(4) Foerster, Jakob, Gregory Farquhar, Triantafyllos Afouras, Nantas Nardelli, and Shimon Whiteson. "Counterfactual multi-agent policy gradients." In *Proceedings of the AAAI Conference on Artificial Intelligence*, vol. 32, no. 1. 2018.

(5) Mnih, Volodymyr, Koray Kavukcuoglu, David Silver, Andrei A. Rusu, Joel Veness, Marc G. Bellemare, Alex Graves et al. "Human-level control through deep reinforcement learning." *nature* 518, no. 7540 (2015): 529-533.

(6) Mnih, Volodymyr, Adria Puigdomenech Badia, Mehdi Mirza, Alex Graves, Timothy Lillicrap, Tim Harley, David Silver, and Koray Kavukcuoglu. "Asynchronous methods for deep reinforcement learning." In *International conference on machine learning*, pp. 1928-1937. PMLR, 2016.

(7) Foerster, Jakob N., Yannis M. Assael, Nando De Freitas, and Shimon Whiteson. "Learning to communicate with deep multi-agent reinforcement learning." *arXiv preprint arXiv:1605.06676* (2016).

(8) Sukhbaatar, Sainbayar, Arthur Szlam, and Rob Fergus. "Learning multiagent communication with backpropagation." *arXiv preprint arXiv:1605.07736* (2016).

(9) Das, Abhishek, Théophile Gervet, Joshua Romoff, Dhruv Batra, Devi Parikh, Mike Rabbat, and Joelle Pineau. "Tarmac: Targeted multi-agent communication." In *International Conference on Machine Learning*, pp. 1538-1546. PMLR, 2019.

(10) Zhang, Sai Qian, Qi Zhang, and Jieyu Lin. "Efficient communication in multi-agent reinforcement learning via variance based control." In *Advances in Neural Information Processing Systems*, 2019.

(11) Kim, Daewoo, Sangwoo Moon, David Hostallero, Wan Ju Kang, Taeyoung Lee, Kyunghwan Son, and Yung Yi. "Learning to schedule communication in multi-agent reinforcement learning." In *International Conference on Learning Representations*, 2019.

(12) Henderson, Peter, Riashat Islam, Philip Bachman, Joelle Pineau, Doina Precup, and David Meger. "Deep reinforcement learning that matters." In *Proceedings of the AAAI Conference on Artificial Intelligence*, vol. 32, no. 1. 2018.

**Additional Feedback:**

Some grammatical mistakes and typos:

- l. 121: "Most MARL problem falls" --> fall
- l. 124: "Other works are aiming at improving" --> Other works are aiming/ aim to improve
- l. 129: "communication is critical cooperative multi-agent problems" --> communication is critical for/ in cooperative multi-agent problems.
- l. 143: "MARL is a rising problem of cooperation and competition among agents" --> Not sure what you want to express here
- l. 152: "those tasks do not conducted in multi-agent setting" --> those tasks do not represent/ are not conducted in a multi-agent setting
- l. 159: "an agent observes an observation and make" --> makes
- l. 163 & l.171: "The episode (...) is considered succeed" --> is considered successful
- l. 164: "the episode is consider" --> considered
- l. 164: "'found' action is sleeted" --> selected
- l. 190: "The Habitat simulator execute the action and return ..." --> executes the action and returns
- l. 191: "sub-environment calculate (...) and send the global reward" --> calculates and sends
- l. 202f: "starting position where the agent starts and the target position where the agent is required to reach" --> target position (which) the agent is required to reach
- l. 204: "We ensures" --> ensure
- l. 246: "This model is implement in the PPO" --> The model is optimised using PPO
- l. 248: "navigaion" --> "navigation"
- l. 249: "The IPPO model (...) share network parameters each agent" --> shares network parameters across all agents
- l. 257: "are shared among agents while the each agent" --> while each agent
- l. 264: "ensure the generazability" --> ensure generalization ?
- l. 286: "trained on by 8 GPUs" --> trained on 8 GPUs
- l. 289: "easily effected" --> easily affected

UPDATE: Author response was recognised and replied to below. Score has not been changed as discussed in response

**Clarity:**

The work suffers from a lack of clarity throughout the paper which largely arise from omitted details and quite a few minor grammatical mistakes or typos. Some found typos/ grammatical mistakes can be found in the additional feedback.

Below, you can find several cases of missing details/ clarity. Addressing such concerns would significantly help to strengthen this work.

- l. 82ff: "In the shared-target navigation sub-task, we mainly evaluate the agents cooperation ability of searching separately for a target." It is unclear to me how separate/ independent navigation of multiple agents requires coordination.
- l. 114ff: "However, our investigation reveals that some of the previous MARL environments, even though claimed to be partially observable, have large observation overlap." Aside from the fully-observable case, there seems to be no immediate correlation between partial observability and observation overlap. Agents could receive large observations with minimal overlap or receive small observations with significant overlap.
- l. 152f: "those tasks do not conducted in multi-agent setting which requires cooperation and communication, and consequently, being more flexible and practical." It is unclear to me how the multi-agent setting requires tasks/ methods to be "more flexible and practical"? Flexible/ practical in what sense?
- l. 191: A "global reward" which is "based on the global state" is mentioned. How is the reward defined? In which regard is this reward function "global"? Is it shared across all agents, all B sub-environments or ...?
- l. 205: The "length of an episode (is constrained) to be between 2m and 20m". Is this length of episodes referring to the shortest path to the goal? The length of an episode would usually be given in the number of timesteps (agent interactions).
- l. 209ff: The average trajectory length for MAIN is stated to be larger than for similar (single-agent) environments. However, it is unclear to me how this "proves that (MAIN) data is more challenging". Long trajectories might still be significantly simpler than shorter trajectories in required action selection if the long trajectory is just a straight line the agent has to navigate whereas the shorter trajectory requires the navigation around obstacles.
- l. 214 & Fig. 3b: A "single-agent navigation task" is mentioned and never elaborated on. Also, it appears this **single-agent** task is executed with varying numbers of agents from 1 to 6. How is this a single-agent task if it apparently does not necessarily have only a single agent in it?
- l. 222: "Distance indicates the average distance forward the target when the agent stops." I suppose the distance to the target when the agent stops is meant.
- l. 225f: Here, samples N are mentioned. What are these samples representing and where are they sampled from? Also, the success indicator Si is not defined. I can only guess this might be a binary indicator for the success of an episode with 1 being success and 0 being failure.
- l. 237: What is $\theta$ the parameter(s) of? The model of one agent/ all agents/ shared?
- l. 242: The random baseline is stated to "randomly sample" actions. Do you mean uniformly random?
- l. 244f: It is stated the baseline model is used to validate if the dataset (among others) has "severe bias". What kind of bias does this refer to and in which way would such a baseline help to identify the bias?
- l. 246ff: It is unclear how the "Multi-single agent" model is trained and applied. It is stated this model "is trained in a single-agent paradigm but tested in a multi-agent paradigm" but this remains vague. Are all agents jointly trained and controlled using a central controller trained with single-agent RL? How is this model applied in a multi-agent setting?
- l. 250: PPO is stated to receive a "single-agent reward" which given the following statement appears to be not influenced by other agents. How is this reward defined?
- l. 257: This is the first time "the CNN and RNN"s are mentioned. Briefly elaborate where these NNs come into the models or refer to a following section where you explain.
- l. 259: The title of this subsection sounds like a task. Make it clear, this is about a proposed communication model.
- Table 2: What does the "Length" metric refer to? Other metrics are outlined in 3.4, but "Length" is never introduced, so it is unclear to which length this corresponds.

**Correctness:**

Throughout the paper, several misleading, contradicting and/or incorrect claims are made:

- l. 27ff (and l.113f): partial observability is no challenge specific to MARL and present within single-agent RL (see POMDP formalism). Also "instability of learning decentralised policies" is a problem, but might be caused by several, different MARL-specific challenges such as non-stationarity and the multi-agent credit assignment problem.
- l. 57f vs l. 160: In l. 57f it is stated that "MAIN does not provide a compass sensor and requires the agent to navigate solely using an egocentric RGB-D camera" which is directly contradicted in l. 160 where it is stated that "the observation contains an RGB-D image, localization information from a GPS compass and contact information from a physics sensor." Is the compass information provided or not?
- l. 66ff vs l. : In l. 66ff It is stated the CTDE framework is "not suitable for real-world simulated environments like MAIN" but in l. 233 it reads that CTDE framework is followed for training in the MAIN environment.
- l. 169: "The task is considered succeed if any agent reaches g and considered failure if any agent fails within its episode." There seems to be a contradiction here. What if some agents within an episode reach their goal g but some fail to do so? Is the episode then considered successful (1st condition fulfilled) or failed (2nd condition also fulfilled)?
- l. 215f: "With the increase of the agent amounts, the difficulty of individual-target task is significantly increasing while the shared-target task is reducing." This appears to contradict the prior statement and/or Fig. 3b. In Fig. 3b it appears that the lengths of shortest paths to the goal location increases for individual target tasks and even more so for shared target tasks with the number of agents. Also, above it is claimed (l. 209ff) that longer paths indicate more challenging tasks. This seems to contradict the statement that with the increasing number of agents (which increases the lengths of shortest according to Fig 3b) the difficulty reduces for the shared-target task?
- l. 231: The MARL learning problem is stated to be formulated as a POMDP. However, POMDPs  formulate single-agent problems. The given citation refers to Olihoeak and Amato (2016)'s book on **Decentralized** POMDPs with the Dec-POMDP formalism which is presumably the intended formalism.
- l. 237f:  The objective which is stated to be minimised is given as the expected episodic return. Presumably, this objective is meant to be maximised.
- General: All MARL approaches introduced in Section 4 are on-policy algorithms based on PPO. However, the MAIN environment introduction in Section 3, in particular Figure 2, suggest that the MARL algorithm model is trained from experience collected in an episodic replay buffer (episode memory). If this replay buffer is applied as in other (MA)RL algorithms, then experiences are staying inside the memory beyond a single optimisation. This would mean that the experience samples used during training are off-policy which violates assumptions made for the optimisation of the on-policy PPO algorithm.

**Documentation:**

For the MAIN environment, provided dataset documentation and benchmark sufficient documentation is largely missing. For the environment, reward functions, observations are not fully outlined. For the dataset, hardly any information is provided regarding the data collection, content of the dataset and its format. For the benchmark, applied tasks and used resources are not fully discussed. See the Weakness section for more details.

However, it should be stated that the authors made efforts to make the dataset accessible and a webpage with some basic instructions as well as code for the benchmark, baselines, methods and environment are all made available. Latter should allow for reproducibility of experiments.

**Ethics:**

There are no ethical concerns I foresee given the content of the dataset and MAIN environment which would require further discussion.



**Relation To Prior Work:**

The environment is sufficiently compared to existing prior work. However, I believe the Deepmind Lab (1) environment (see weakness section) should be mentioned.

The overview over MARL provided in Section 2 appears to cover a selective view of MARL research. A more extensive overview and coverage in the benchmark would significantly strengthen any claims.

However, the main aspect in which prior work seems to be ignored is communication in MARL. Given a novel communication model is proposed in 4.3, I would expect a comparison to prior work and explanation how the new approach differs from these related works. See references (7 - 11) in weakness section for some prominent examples.



**Summary And Contributions:**

This paper makes four contributions all focused on multi-agent reinforcement learning (MARL) using 3D egocentric visual inputs for close-to-real-world observations:

1. The Multi-Agent Indoor Navigation (MAIN) environment is proposed and made available which aims to represent a MARL environment which requires significant coordination and  "realistic first-person" visual observations.
2. A large dataset with episodic data is made available for learning and testing.
3. A benchmark of several MARL baselines in MAIN environment tasks is conducted.
4. A communication module is proposed which is found to benefit MARL performance in MAIN.

---

> ### Author Response · Authors · 2021-07-14
> **Response**
>
> Thank you for your thorough review and so many helpful suggestions. And thank you for recognizing our contributions on observation overlap, the convincing challenge and nice visualization of the difficult MAIN task.
>
> **Motivation:**
>
> Even though we build our environment upon the Habitat simulator, it is more realistic compared to other simulators, like MARLÖ and Deepmind Lab. The Habitat simulator has more realistic visual observation that is reconstructed from a panoramic camera, however, the MARLÖ and the Deepmind Lab use computer graphics to render the visual observation, which is far from the real-world. The Habitat simulator is also equipped with a more complex physical engine, Bullet (https://pybullet.org/wordpress/), which is more realistic compared to game engines. Above all, although we build our environment upon a simulator, our benchmark is a step toward real-world applications.
>
> **MAIN environment:**
>
> The RGB-D resolution is 256*256, the reward function is defined by the shortened distance of the minimal distance between agents and the target. The turn action makes an agent turn for 10 degrees and the forward action makes the agent move forward for 0.25m. We will make the details clearer in the revision.
>
> Our environment reconstructs a 3D house and renders observations for agents for each step, which is computation costly. Other MARL simulators like SMAC [1] and Hide-and-seek [2] get observations directly from interfaces with few data processing. However, this formatted data with few noise is far from the real-world. By our proposed asynchronous-synchronous pipeline, the simulation speed retrains the same as single-agent Habitat (1000 FPS for 512*512 RGB+depth). If we simulate multiple agents in a process like iGibson simulator [3] rather than using the asynchronous-synchronous pipeline, the simulation speed will slow down to about 200 FPS.
>
> Large-scale dataset:
>
> 1. The dataset contains the starting position of an agent and the target position, and the target class id.
> 2. The data is formatted in a tree-structure and stored in a json file.
> 3. We will provide an example of our data on our website and improve our documentation.
> 4. The data collection method is mentioned in Section 3.3. We use the room textures and other 3D assets provided by Matterport3D to build the MAIN environment. The data for training and testing are episodes. Each episode is defined by a starting position where the agent starts and the target position which are randomly sampled from navigable points
>
> **Communication module**
>
> 1. We will introduce the related work and describe more on our motivation for the design of the communication module.
> 2. The  "receiver" or "sender" are not predefined. All agents are both receiver and sender since they swap their hidden states for each step. Thus, the communication module is a symmetric structure.
> 3. Actually, the variant No. 1 and No. 2 are the practice of previous works [4, 5]. And we find that communication by recurrent state rather than value is helpful.
>
> [1] M. Samvelyan, T. Rashid, C. S. De Witt, G. Farquhar, N. Nardelli, T. G. Rudner, C.-M. Hung, P. H. Torr, J. Foerster, and S. Whiteson. The starcraft multi-agent challenge
>
> [2] B. Baker, I. Kanitscheider, T. Markov, Y. Wu, G. Powell, B. McGrew, and I. Mordatch. Emergent tool use from multi-agent auto curricula
>
> [3] F. Xia, A. R. Zamir, Z. He, A. Sax, J. Malik, and S. Savarese. Gibson env: Real-world perception for embodied agents
>
> [4] R. Lowe, Y. Wu, A. Tamar, J. Harb, O. P. Abbeel, and I. Mordatch. Multi-agent actor-critic for mixed cooperative-competitive environments.
>
> [5] C. Yu, A. Velu, E. Vinitsky, Y. Wang, A. M. Bayen, and Y. Wu. The surprising effectiveness of MAPPO in cooperative, multi-agent games

---

### Official Review · Reviewer_SKNX · 2021-07-04
**An incremental contribution on top of Habitat**

**Rating:** 4
**Confidence:** 3

**Strengths:**

1. The first multi-agent environment with photorealistic 3D simulation.
2. Baseline algorithms are provided.

**Weaknesses:**

1. The paper is an incremental update of Habitat. It extends an existing environment with multi-agent reinforcement learning algorithms. Most benefits of this task are inherited from Habitat.
2. The experiments do not provide new insight into these algorithms.
3. No multiple runs and error bars, which are a must-have for unstable RL algorithms.
4. No mention of the simulation speed (e.g., How does it compare to single-agent simulation? How does it compare to other environments?)

**Additional Feedback:**

Some minor typos:
1. L19 "By Experimenting ..." -> "By experimenting ..."
2. L237 "We optimization ..." -> "We optimize ..."


**Clarity:**

The paper is well written with figures that are very helpful for understanding.
But there are plenty of typos in the paper.

**Correctness:**

The claims are correct and the techniques are sound.
The experiment design follows common practice and the results are reproducible.

**Documentation:**

The simulator has basic documentation on training and evaluating the agents.
The hosting license is mentioned in the checklist but not in the code repository or webpage.
But comprehensive instructions to reproduce all experiments in the paper are missing.

**Ethics:**

No.

**Relation To Prior Work:**

This paper clearly discussed its relation to prior work in Section 2 and Table 1.

**Summary And Contributions:**

This paper proposes "MAIN": a multi-agent reinforcement learning benchmark for indoor navigation.

It makes the following contributions:
1. Extend the habitat environment to support multi-agent and collect a large-scale dataset.
2. Benchmark several baselines algorithms on this task.
3. Propose a new cooperative communication mechanism.

---

> ### Author Response · Authors · 2021-07-14
> **Response**
>
> 1. We propose a novel and challenging multi-agent navigation benchmark. Compared with Habitat that is able to research on single-agent navigation only, the MAIN benchmark is able to research on the communication problem between multiple navigation agents, which is a critical problem in MARL. Compared with previous MARL environments, our environment provides a realistic embodied observation with low observation overlap, which is novel and challenging. A more detailed comparison is shown in Table 1.
> 2. Our experiments validate the challenge that communication is the key to the multi-agent tasks when agents share few observation overlap.
> 3. We will provide multiple runs with error bars. The curves in Figure 5 show that our training curves are stable.
> 4. By our proposed asynchronous-synchronous pipeline, the simulation speed retrains the same as single-agent Habitat (1000 FPS for 512*512 RGB+depth). If we simulate multiple agents in a process like iGibson, the simulation speed will slow down to about 200 FPS.

---

### Official Review · Reviewer_mazx · 2021-07-06
**A new challenging MARL benchmark built on the Habitat simulator**

**Rating:** 7
**Confidence:** 3

**Strengths:**

The new benchmark appears challenging, and the experiments show it is possible to solve.
It appears to be reasonably easy to use.
The provided diagrams are very helpful for understanding the architectures used in the experiments as well as details about the environment.

**Weaknesses:**

The benchmark appears to be mostly distinguished from prior benchmarks by having a challenging observation space.
This may decrease interest in the benchmark, since learning navigation from egocentric observations is already a challenging research topic.
The experiments use only PPO and IPPO with a selection of different neural network architectures.
Although these experiments show the benchmark is achievable and give an approximate sense of difficulty, there is no statistical analysis or other interpretation of the results.
It is also unclear how difficult hyper-parameter tuning is for this benchmark.

**Additional Feedback:**

The authors should add a license file to their code repository and provide some documentation on how to use the environment without the provided runner.

**Clarity:**

The paper is acceptably written. There are several grammatical errors, but they do not make the paper difficult to read.

**Correctness:**

The evaluation and experiment methods appear to be appropriate, although somewhat minimal.

**Documentation:**

The documentation is fairly clear on how to install and run the benchmark. It could use more detail on how to use it other algorithms than those provided.
The code appears to be licensed under the MIT license, however a license file is missing from the source repo.
There appears to some plan for hosting, and there is appears to be sufficient detail to reproduce the results.

**Ethics:**

There do not appear to be any ethical concerns that warrant further discussion or review, since this benchmark only uses data that was already use in earlier works (namely, Habitat and MultiON).

**Relation To Prior Work:**

It is clear how this work differs from and builds on previous contributions. It also includes a useful table comparing different existing MARL benchmarks.

**Summary And Contributions:**

This submission introduces a new MARL benchmark built on the Habitat simulator.
It also provides a few experiments showing the difficulty of the new benchmark.

---

> ### Author Response · Authors · 2021-07-14
> **Response**
>
> Thank you for recognizing our benchmark is challenging, our framework is easy to use and our diagrams are helpful for understanding.
>
> 1. In addition to the difficulty of the observation space, our benchmark is more challenging on transition function. Previous environments receive the values of position coordinates as an observation and these values are updated linearly when agents are moving while in the embodied view, the observed change is highly non-linearity.
> 2. In Fig. 5, we analyze how the structures of communication impact the learning curves. We will introduce more analytical experiments on more MARL algorithm backbones like QMIX and COMA in the revision.
> 3. Most hyper-parameter settings inside a single model are as follows [1]. The hyper-parameters for MARL are demonstrated in the Implementation details in Section 5.
> 4. We will provide more metrics to better evaluate the performance of multi-agent navigation.
> 5. We will fix the grammatical errors in the revision.
>
> [1] Saim Wani, Shivansh Patel, Unnat Jain, Angel X. Chang, Manolis Savva, MultiON: Benchmarking Semantic Map Memory using Multi-Object Navigation, NeurIPS 2020

---

> > ### Comment · Reviewer_mazx · 2021-07-20
> > **Thank you**
> >
> > I believe this paper will be a valuable contribution if the above improvements are made, and I've updated my review appropriately.

---

### Decision · Program_Chairs · 2021-07-26

**Decision:**

Reject

**Comment:**

This paper introduced MAIN, a multi-agent indoor navigation benchmark built upon the Habitat simulator. The three expert reviewers have expressed several similar concerns at the initial reviews, including the incremental contributions over Habitat and other existing multi-agent benchmarks and the lack of clarity/details of the experiments. In particular, reviewer YYCW has written a comprehensive review of this paper and pointed out several major weak points of this submission. The authors' responses did not sway this reviewer's opinion, who voted Clear Reject in the end. The AC read the paper, the reviews, and the post-review discussions carefully and concurred with the reviewers' overall assessment of this paper. While this paper has made a promising first step towards studying multi-agent visual navigation in more realistic environments, it is unclear that the simple combination of multi-agent reinforcement learning and visual navigation would give rise to more fundamental challenges than previous efforts that studied these two problems separately but in a more systematic manner. Furthermore, the evaluation results are preliminary and lack statistical analysis, making it difficult to assess the technical difficulties of the proposed tasks. The writing quality could also be further improved with more details about the simulation environments and the learning models, as requested by the reviewers. Taking all into account, the AC recommended rejecting this work.